# Active site localization of methane oxidation on Pt nanocrystals

Dongjin Kim[1], Myungwoo Chung[1], Jerome Carnis [1], Sungwon Kim[1], Kyuseok Yun[1], Jinback Kang[1], Wonsuk Cha [2,3], Mathew J. Cherukara[3], Evan Maxey[3], Ross Harder[3], Kiran Sasikumar[4], Subramanian K. R. S. Sankaranarayanan[4], Alexey Zozulya[5], Michael Sprung[5], Dohhyung Riu [6] & Hyunjung Kim [1]

High catalytic efficiency in metal nanocatalysts is attributed to large surface area to volume ratios and an abundance of under-coordinated atoms that can decrease kinetic barriers. Although overall shape or size changes of nanocatalysts have been observed as a result of catalytic processes, structural changes at low-coordination sites such as edges, remain poorly understood. Here, we report high-lattice distortion at edges of Pt nanocrystals during heterogeneous catalytic methane oxidation based on in situ 3D Bragg coherent X-ray diffraction imaging. We directly observe contraction at edges owing to adsorption of oxygen. This strain increases during methane oxidation and it returns to the original state after completing the reaction process. The results are in good agreement with finite element models that incorporate forces, as determined by reactive molecular dynamics simulations. Reaction mechanisms obtained from in situ strain imaging thus provide important insights for improving catalysts and designing future nanostructured catalytic materials.

[1] Department of Physics, Sogang University, Seoul 04107, Korea. [2] Materials Science Division, Argonne National Laboratory, Argonne, IL 60439, USA. [3] Advanced Photon Source, Argonne National Laboratory, Argonne, IL 60439, USA. [4] Center for Nanoscale Materials, Nanoscale Science and Technology Division, Argonne National Laboratory, Argonne, IL 60439, USA. [5] PETRA III, Deutsches Elektronen-Synchrotron (DESY), D-22607 Hamburg, Germany. [6] Department of Materials Science and Engineering, Seoul National University of Science and Technology, Seoul 01811, Korea. Correspondence and requests for materials should be addressed to H.K. (email: hkim@sogang.ac.kr)

Metal nanoparticle-based heterogeneous catalysts play an important role in energy conversion and environmental technologies[1,2]. In addition to their larger surface to volume ratio, they have highly under-coordinated sites. In general, under-coordinated sites including steps and kinks can enhance the catalytic efficiency[3–6] by decreasing kinetic barriers of dissociation and/or bond breaking of the reactants[7]. Interactions between reactants and catalysts are a key factor determining reactivity[8,9] in heterogeneous catalysis. Since structure and interaction are coupled under reaction conditions, several studies reported overall shape or size changes of metal nanoparticles[10–13] during catalytic processes. They observed facet rearrangement, contraction and relaxation of the bond distance, and surface morphological refacetting, and so on.

In heterogeneous catalysis, the identification of active sites[14] is important in terms of understanding the catalytic mechanism and improving the efficiency of the surface catalytic reaction. Active sites have been identified as low-coordinated atoms by microscopy techniques such as scanning tunneling microscopy[14] and scanning transmission electron microscopy[15]. Infrared spectroscopy is sensitive to interactions between catalysts and reactants in terms of chemical adsorption[16] and X-ray absorption spectroscopy is useful for investigating the electronic states related to chemical bonding in the catalysts[17]. However, the structural changes at the active sites have not been clearly understood so far.

Bragg coherent diffraction imaging (BCDI) is sensitive to distortions of the crystal lattice, owing to the highly sensitive nature of this technique to minor modifications of the phase of a scattered wave field[18–20]. This is accomplished through careful measurements of coherent X-ray diffraction (CXD) patterns in the vicinity of Bragg peaks of the sample and computational methods[21,22] to retrieve both the shape of the nanocrystals and the internal displacement from the relative phases of the interference pattern.

Here we study distortions at active sites in platinum (Pt) nanocrystals using in situ BCDI, above and below the activation temperature of catalytic methane oxidation[23] as an example of catalytic process. The results show the severe crystal lattice displacement at the edges than the facets due to the adsorption of oxygen and catalytic reaction of methane. In addition, the simulations by reactive molecular dynamics (RMD) informed finite element analysis (FEA) are consistent with 3D reconstructed images from the CXD patterns.

## Results

**In situ structural evolution during the catalytic process.** Pt nanocrystals were synthesized by dewetting from a thin Pt film deposited on $Al_2O_3$ substrates. (Supplementary Figure 1 shows a typical SEM image.) The study was conducted in stages, with 3D CXD measurements performed at the following steps: (1) oxygen adsorption and (2) methane oxidation. Figure 1 schematically shows the BCDI experimental setup and CXD patterns measured at the (111) Bragg peak of an individual Pt nanocrystal above the activation temperature. Here we selected the size of the Pt nanocrystals to be ~200 nm to provide a distinct deformation field distribution in a single nanocrystal within the typical spatial resolution of BCDI (~14 nm). The detailed experimental conditions regarding gas environment are described in Methods. The initial CXD pattern in $H_2$ (Fig. 1a) showed the sixfold symmetry expected of a Pt nanocrystal with a truncated octahedral shape, with (111) and (100) facets. As the oxygen adsorption progressed (Fig. 1b, c), the center of the Bragg peak changed shape from circular to triangular, and the fringe pattern became strongly asymmetric. In particular, the first fringe showed noticeable changes, indicating that the Pt nanocrystal deformed almost through the entirety of its volume[24]. The maximum distortion was observed in Fig. 1d when $CH_4$ was introduced. The distortion

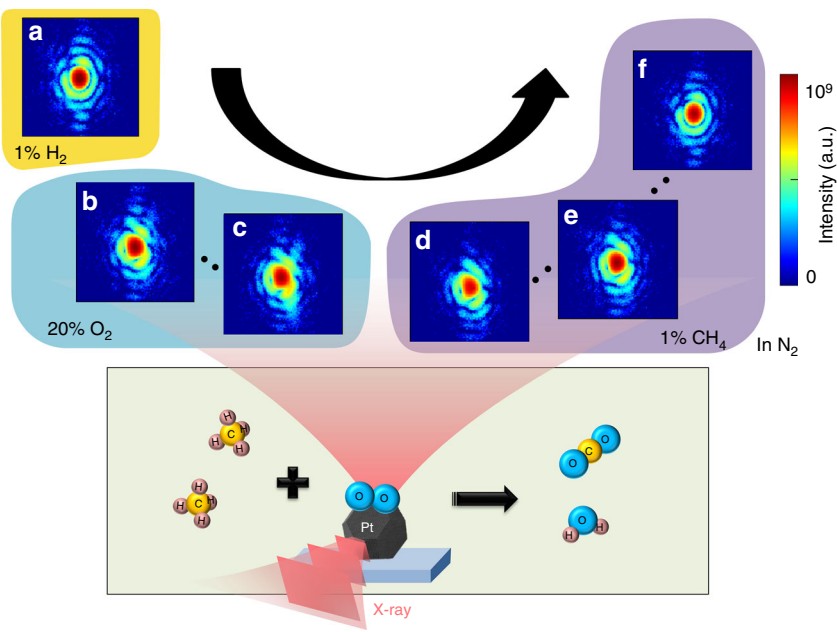

**Fig. 1** Schematic of in situ Bragg coherent X-ray diffraction imaging (BCDI). The BCDI measurement scheme during methane catalytic oxidation illustrates the acquisition of (111) Bragg coherent diffraction patterns from the same Pt nanocrystal throughout. Slices through the Bragg coherent diffraction patterns are measured for Pt nanocrystals above the catalytic activation temperature in the presence of different gases. **a** After 36 min in $H_2$, **b** 2.6 min, and **c** 36 min after the $O_2$ insertion, **d** ~2.6 min, **e** 16 min, and **f** 36 min after $CH_4$ was introduced. Under a 20% $O_2$ gas flow, the diffraction pattern becomes distorted (**b** and **c**), and continue to evolve when $CH_4$ is added. The distortion fades in **f** owing to completion of the methane catalytic oxidation process in 1% $CH_4$

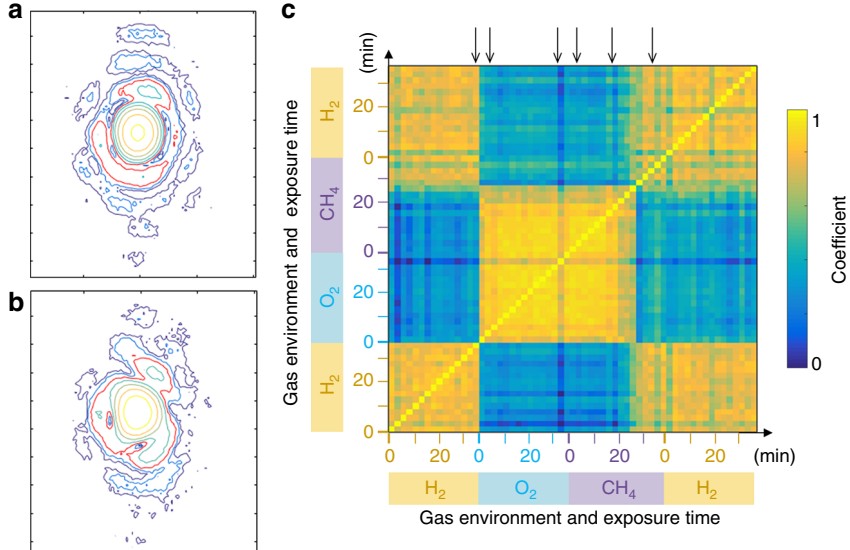

**Fig. 2** The cross-correlation map of the coherent X-ray diffraction (CXD) patterns. **a, b** Contour of intensities for the CXD pattern shown in Fig. 1a and Fig. 1b, respectively, with the red contour identifying the first fringes. **c** The cross-correlation map between each CXD pattern above activation temperature as a function of different gas flow conditions. The areas used in the correlation analysis are indicated as dashed boxes in the Supplementary Fig. 3a, b. Gas environments together with the exposure times are indicated on the x and y axis. Black arrows on the top show the time and gas environment for Fig. 1a–f. Time 0 indicates the start of each gas flow (1% $H_2$, 20% $O_2$, 1% $CH_4$, and 1% $H_2$). The correlation coefficient, 1, means total positive linear correlation and 0 no linear correlation

continued in Fig. 1e throughout the chemical reaction until the catalytic process completed. Strain release was evident in the CXD pattern in Fig. 1f. For comparison, the patterns recorded below the activation temperature did not show any notable changes (Supplementary Fig. 2).

To understand how distortion evolved during the catalytic process, the Pearson correlation function[25] was applied to the total three-dimensional (3D) CXD patterns. The Pearson correlation coefficient for CXD patterns is

$$\rho(A, B) = \frac{1}{N-1} \sum_{i=1}^{N} \left( \frac{A_i - \bar{A}}{\sigma_A} \right) \left( \frac{B_i - \bar{B}}{\sigma_B} \right),  \quad (1)$$

where $A_i$ and $B_i$ are the scattering intensity values for a particular pixel in the pattern, $\bar{A}$ and $\bar{B}$ the mean values of the scattering intensities, and $\sigma_A$ and $\sigma_B$ the standard deviations, for the two CXD patterns, A and B. The sum is evaluated over 3D CXD patterns and $\rho(A,B)$ is computed for the selected area around the first fringe where shows maximum distortion. (See Supplementary Fig. 3.) Fig. 2a, b show contour slices through the CXD patterns (Fig. 1a, b, respectively) with the red contour identifying the first fringes. The cross-correlation map in Fig. 2c was computed for the CXD patterns measured in various gas environments by using Eq. 1, and detailed procedure is described in Methods. Gas environments together with the exposure times are indicated on the x and y axis, and the experimental conditions for Fig. 1a–f are also indicated on the map with black arrows. Time 0 indicates the start of each gas flow. The correlation coefficient, 1 means total positive linear correlation, 0 means no linear correlation, and −1 means total negative linear correlation. Immediate distortion appeared after oxygen gas was introduced. The original diffraction pattern was recovered after ~22 min in $CH_4$. This reversion can be attributed to the available oxygen being consumed by catalytic oxidation. Gas analyses were carried out to check the catalytic reaction and resultant products ($CO_2$ and $H_2O$).

**Reactive molecular dynamics simulation.** To elucidate the reaction mechanism of methane oxidation on Pt, large-scale RMD simulations[26,27] were performed. Snapshots of the atomic displacements of Pt nanocrystals in $H_2$, in $O_2$, and in $CH_4$ after oxidation, are shown in Fig. 3a–c, respectively. In Fig. 3a, we observed limited interactions between Pt and H, consistent with the experimental observations. Figure 3b shows the effects of oxidation of the nanocrystal at edge and corner sites, where Pt atoms are under-coordinated, after exposure to $O_2$ for 100 ps. Molecular oxygen also physisorbed on the surfaces of the Pt nanocrystal. To simulate methane oxidation in limited adsorbed oxygen environment, we removed un-adsorbed $O_2$ molecules, while adsorbed $O_2$ molecules and bonded O atoms were retained. We then introduced $CH_4$ molecules and ran additional isothermal-isobaric dynamics for 45 ps. Figure 3c shows the structure of the particle after 44 ps. We observed the bonding of $CH_4$ molecules to dissociated O atoms at edge and corner sites (shown in Supplementary Fig. 4). However, we did not observe the final products escaping the surface of the nanocrystal within the short timescales of our RMD simulations[28]. Notably, only a fraction of adsorbed oxygen atoms was an active site for methane oxidation owing to steric hindrance. Thus, we calculated the ratio of adsorbed methane to oxygen atoms ($CH_4/O$) on the Pt nanocrystal in the RMD simulation in Fig. 3d. The $CH_4/O$ adsorption ratio was fitted to an inverse exponential, which was extrapolated to infinite time to yield an adsorption ratio of 0.398 ± 0.003. Although the kinetics at picosecond timescales of atomic level processes are difficult to translate into experimental particle sizes, the resultant values of local atomic displacements are expected to be similar for larger Pt nanocrystals.

**3D imaging and RMD-informed finite element analysis.** The atomic displacements obtained from the measured CXD patterns by phase retrieval process are compared with the calculated results in Fig. 4. Figure 4a–d show 3D images and cross-sectional images from the CXD patterns shown in Fig. 1a, b, d, f,

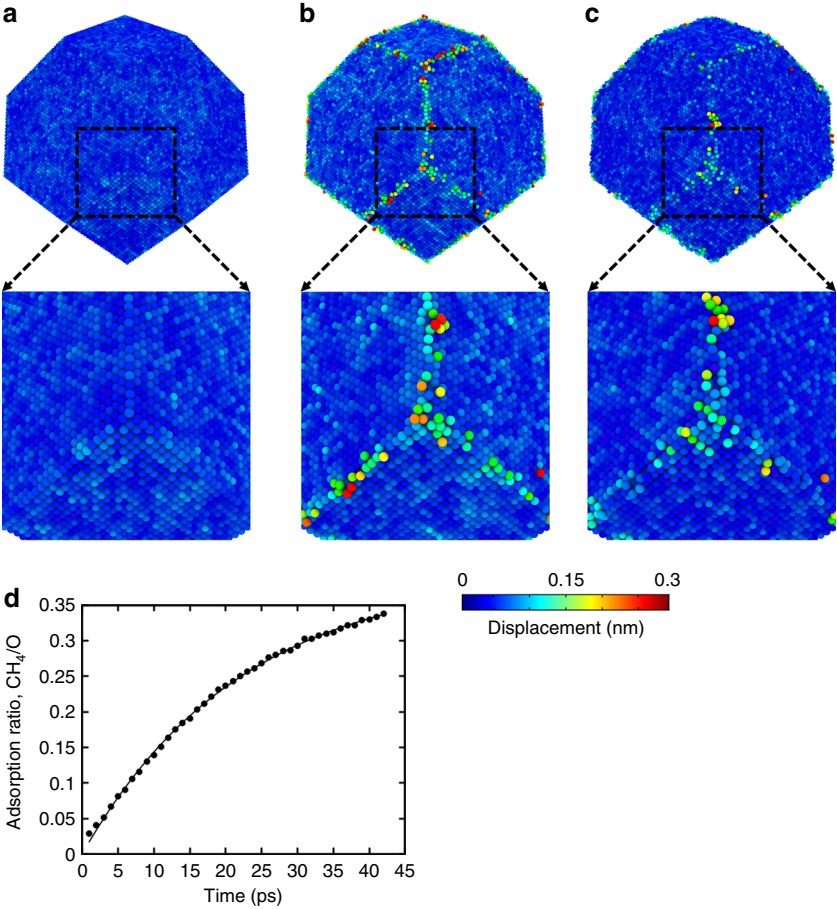

**Fig. 3** Reactive molecular dynamics (RMD) simulation of Pt catalytic activity. Images of a Pt nanocrystal colored by displacement from RMD simulation results and expanded views of the region marked out in the upper image. **a** Pt in an environment of pure $H_2$. **b** Pt in an environment of pure $O_2$. Surficial Pt atoms are displaced from their original position by oxidation of the edge and corner sites; those on faces of the particle show only weak displacements. **c** Oxidized Pt nanocrystals in the presence of $CH_4$. **d** Analysis of the RMD of the ratio of adsorbed methane molecules to adsorbed oxygen atoms on the Pt nanocrystal. Only a fraction of the absorbed oxygen atoms are active for final product formation owing to steric hindrance. The solid line is a fit to $A \times [1 - \exp(-t/B)]$ with $t$ in ps, $A$ and $B$ are fitting constants. Extrapolating to infinite time yields an adsorption ratio of 0.398 ± 0.003 at any given time

respectively. 3D images are colored by the local displacement field at the surface and at vertical cross-sections with reference to a perfect Pt lattice. Red (positive sign) indicates the projected displacements along the [111] direction, $u_{111}$, and blue (negative sign) implies the opposite direction. Note all the displacements shown here are after the refractive index correction[29]. (See Supplementary Fig. 5.) The residual distortion in the as-synthesized crystal in Fig. 4a is a maximum value of $u_{111}$ as 7% of lattice constant of $Pt_{111}$. Figure 4b shows strong contraction at the edge due to chemisorbed O atoms from dissociation of $O_2$ molecules with maximum $u_{111}$ as 20% of the lattice constant. Because the direction of the projected displacements starting at the edge faces toward the interior of the Pt nanocrystal, this indicates contraction. (Detailed interpretation can be found in Supplementary Fig. 6.) Different projections of the displacement field arisen from the edges can be measured at other Bragg peaks[30]. In Fig. 4c, the greatest distortion is detected because of the bonding of $CH_4$ molecules to dissociated O atoms. The maximum $u_{111}$ value is ~30% of the lattice constant. The cross-sectional images in Fig. 4b, c clearly demonstrate that contraction originated at the edges is propagated to the interior of the crystal. As shown in Fig. 4d, after methane oxidation ended, the contracted lattice was released. (See Supplementary Fig. 7 for expanded time range.)

The CXD image of the nanocrystal was imported into an RMD-informed finite element model, and suitably meshed. We emphasize that FEA was performed on the density of the CXD image with suitable substrate constraints and input forces estimated from the RMD simulation, as shown in Fig. 4e. The force exerted by an adsorbed O atom on a Pt atom was determined from the RMD simulation to be $1.17 \times 10^{-10}$ N. Assuming 14.8% coverage of oxygen (see Methods) on the edge sites at the top of the nanocrystal (having 14.99 Pt atoms per $nm^2$), the total compressive stress $P_{MD}$ was determined to be 259.74 MPa. The steady-state FEA solution for this system revealed a maximum $u_{111}$ of 0.04 nm (Fig. 4g). Further adsorption of methane at active oxygen sites resulted in an additional compressive force on a Pt atom of $1.12 \times 10^{-10}$ N. With an adsorption ratio of 0.398 and methyl coverage of 5.88%, an additional compressive stress of 98.79 MPa is applied at the edge sites on the top of the nanocrystal. Figure 4h shows steady-state FEA solutions resulted in a maximum $u_{111}$ of 0.07 nm (consistent with the experimental observations in Fig. 4c). In addition, the steady-state displacement field extended ~50 nm into the particle interior. The similarity of the FEA to the BCDI results confirmed the RMD simulations and indicated that the under-coordinated corners and edges of the Pt nanocrystal acted as preferential adsorption sites for oxygen.

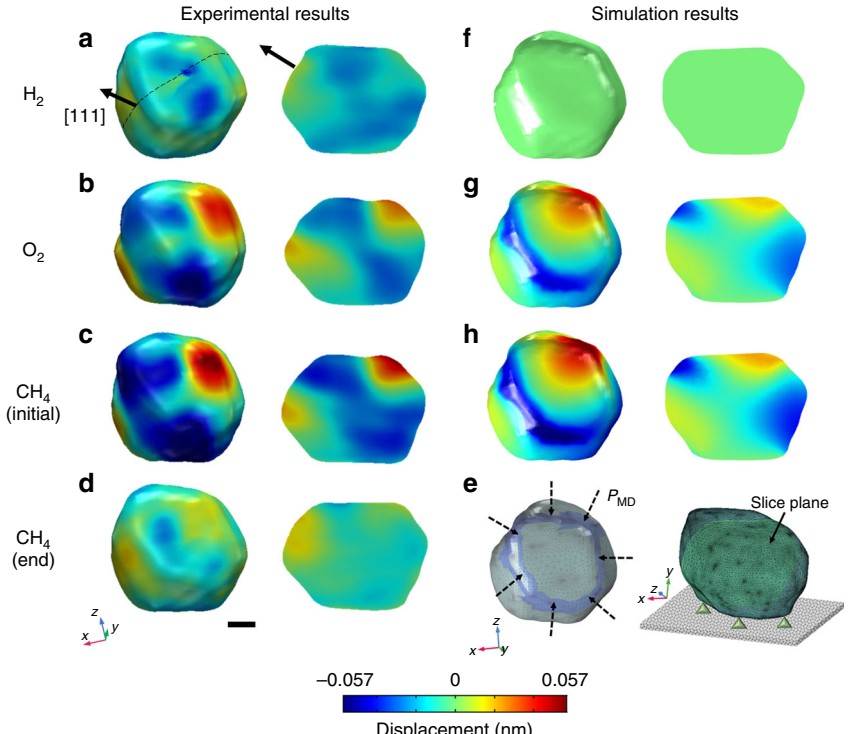

**Fig. 4** Displacement distribution from the experiments and finite element analysis (FEA). **a–d** 3D reconstructed images (left) and sliced images (right) including lattice displacements at (111) with 25% isosurface amplitude of a 220 nm Pt nanocrystal under each of the four gas environments for Fig. 1a, b, d, f, respectively. Red (positive sign) indicates the projected displacements along the [111] direction and blue (negative sign) implies the opposite direction. These 3D images show lattice contraction along the edges and corners owing to adsorption of oxygen atoms and methane oxidation. The cross-sections show that the distortion propagates deep into the interior of the Pt nanocrystal and is released after catalytic methane oxidation. **e** Images on the left show meshing of BCDI result. RMD-derived reaction-induced pressure ($P_{MD}$) is applied to the region shaded in blue. Images on the right show the Dirichlet constraint of zero displacement, included in the FEA. The slice plane used for **f–h** is also shown. **f–h** The FEA predictions for the Pt nanocrystal response to adsorption induced external force obtained from RMD corresponding to **a–c**. Crystal and slice planes are colored by the [111] projected displacement. Scale bar corresponds to 50 nm

## Discussion

We note that there were not enough final product formation events on the short RMD timescale to achieve a statistically reliable estimate of the direction of the force (compression vs. tension) owing to methane adsorption. The BCDI experimental reconstructions clearly indicated that addition of methane induced compression of Pt. Hence, we assumed a compressive force in the FEA simulation. The obtained match in displacement magnitudes and distributions within the crystal between RMD-informed FEA and BCDI is a remarkable result. This finding, (1) validates the hypothesis that adsorption events on Pt edge sites cause the observed displacement field. (2) Furthermore, it is likely that the top edge sites of the Pt crystal are most active. (3) The simulation spanned several length scales and remained consistent with the experimental observations, emphasizing the strength of this integrated approach to investigating nanocatalysis behavior.

In addition, to estimate exact strain field distribution inside the crystal, the strain $(\partial u_{111}/\partial x_{111})$[25] was calculated for 3D images in Fig. 4. Supplementary Figure 8a–d show the 3D images in Fig. 4a–d with front view of (100) facet (on the left) and vertical slices (on the right). The highly strained area ($|\partial u_{111}/\partial x_{111}| > 2 \times 10^{-3}$) in $O_2$ locates at the corner of (100) facet with tensile strain (in red, positive values) on the left and compressive strain (in blue, negative values) on the right, and the area enlarges with introduction of $CH_4$. This anisotropic strain might be clue for shape changes observed during general oxidation processes[10–13].

In conclusion, using in situ BCDI, we observed deformation of a single Pt nanocrystal with a considerable compressive strain

starting at the edge indicating localization of active sites during catalytic methane oxidation. Contraction at the edges arises due to dissociated O atoms and the deformation becomes stronger when $CH_4$ molecules bond to the dissociated O atoms during the chemical reaction with $CH_4$. The deformation field distribution arising from the interaction forces between the reactants and the Pt catalyst was confirmed by RMD-informed FEA. Our work identified the active sites underlying atomic-scale catalytic activity, where it can induce structural changes of nanocrystals during the catalysis.

## Methods

**Sample preparation**. Pt nanocrystals were prepared by dewetting from Pt films for 14 h at 1700 °C under argon flow in a tube furnace (GSL-1750X, MTI Corporation). Pt films were deposited on sapphire (0001) substrate with an electron beam evaporator. A typical scanning electron microscope (SEM) image of a sample is shown in Supplementary Fig. 1. SEM measurements were performed with a scanning electron microscope SNE-4500M (SEC).

**Bragg coherent X-ray diffraction imaging experiments**. Focused coherent X-rays with a wavelength of 0.1377 nm from a Kirkpatrick–Baez mirror at the 34-ID-C beamline in Advanced Photon Source, USA, illuminated isolated samples mounted in a sample chamber under gas flows of 1% $H_2$, 20% $O_2$, and 1% $CH_4$ with a $N_2$ base gas flow of 50 mL per min over various temperatures. Before and after the catalytic processes, the CXD patterns were recorded in a flow of 1% $H_2$ in $N_2$ for ~30 min to clean the Pt surface. The samples were exposed to a flow of 20% $O_2$ in either He or $N_2$ for oxygen adsorption and to a flow of 1% $CH_4$ in either He or $N_2$ for methane oxidation. Before introducing $CH_4$ into the sample chamber, the remaining $O_2$ gas was completely removed by evacuation. The CXD patterns were measured with a Timepix detector (Amsterdam Scientific Instruments) with 55 × 55 μm² pixel size located ~0.65 m away from the sample. Three-dimensional

diffraction data were collected as rocking curves of the sample tilt angle in steps of 0.01° with a total of 61 frames.

At the P10 beamline in PETRA III, Germany, coherent X-rays with a wavelength of 0.1486 nm were focused by compound refractive lenses to illuminate the Pt nanocrystal samples. The samples were mounted in a sample chamber under a gas flow of 1% $H_2$, 20% $O_2$, and 1% $CH_4$ in a He base gas flow at 100 mL per min with varying temperature. The gas exchange process was same as above. CXD patterns were measured with a Lambda detector (X-Spectrum) with $55 \times 55$ μm² pixel size located ~1.83 m away from the sample. Three-dimensional diffraction data were collected as rocking curves of the sample tilt angle in steps of 0.01° with a total of 61 frames.

**Phase retrieval algorithm**. We used a phase retrieval package that has been used extensively[31]. After loading CXD patterns from BCDI measurements with a random phase, 620 iterations with constraints consisting of error reduction and difference map (hybrid input-output) combinations were progressed with guided analysis (GA). The GA method helps to select the best reconstruction from the lowest sharpness metric, which is the summation of the absolute value raised to the fourth power for the reconstruction result. This process was performed for five independent reconstructions with three generations, where the previously used reconstruction was set as the seed for the next one.

**Mass spectroscopy measurements**. The gas analysis was carried out with a mass spectroscope (Prisma Plus™ QMG220) and mass flow controllers (MFCs). To measure the final products with enough S/N, the Pt powder (200 nm) in capillary was used. Below and above the activation temperature, 20% $O_2$ gas in a $N_2$ base gas was flowed at 50 mL per min with a MFC for 40 min, and then flushed out. After flushing, the sample was exposed to a 1% $CH_4$ gas flow in a $N_2$ base gas flowed at 50 mL per min with a MFC for 40 min. $CD_4$ gas was also used for identifying the final products.

**Cross-correlation analysis**. Cross-correlation analysis was applied for observing changes in the CXD patterns in various gas environments with time. The cross-correlation map was obtained by the Pearson correlation function, which calculates the covariance of two variables divided by the product of their standard deviations. We focused on the first fringe region of the CXD pattern in Fig. 2a and Fig. 2b (selected areas marked as dashed boxes in the Supplementary Fig. 3a, b) because the region showed maximized distortion generated by strain throughout nearly the whole volume of the Pt nanocrystal[25]. Note that the selected regions were not limited to the two-dimensional areas but to the 3D volume in the integrated 3D CXD patterns. The cross-correlation maps for the four selected regions ①, ②, ③, and ④ are displayed in the Supplementary Fig. 3c–f, respectively. The correlation coefficient implies the average of the normalized intensity correlation at each pixel position in the selected region.

**Reactive molecular dynamics simulation**. Reactive molecular dynamics simulations were performed with the use of the idealized Pt nanocrystals (~10 nm diameter, 250,000 atoms) created in the shape of a truncated octahedron. All simulations were performed within the reactive force field framework with the potentials published by Singh et al.[27] The Pt nanocrystal was equilibrated at 0 K through an energy minimization before being heated up to 300 K in 30 ps. The particle was subsequently heated to 600 K in the presence of $H_2$ in 30 ps and then held at that temperature for 100 ps. The same procedure was used for the particles in an oxygen environment. The oxidized Pt structure was then placed in a $CH_4$ environment at 600 K and dynamics were run for 45 ps. All simulations were performed under isothermal-isobaric conditions using the Nose–Hoover thermostat and barostat as implemented in the LAMMPS package[28]. Equations of motion were integrated over a timestep of 0.1 fs by the verlet integrator. Atomic structures were rendered by the Ovito package[32].

**RMD-informed finite element analysis**. The FEA calculations were performed in COMSOL with the stationary structural mechanics module, while the RMD simulations were performed in LAMMPS with the ReaxFF interatomic potential. The FEA mesh contained elements in the size range of 1–15 nm. We found that the number of total adsorbed oxygen atoms (averaged over 100 ps of RMD simulation) on the active edge sites was 2820. The simulated nanocrystal had a total surface area of 1271.13 nm², yielding an oxygen adsorption activity per unit area of 2.22 atoms per nm². On a (111) surface Pt edge site (14.99 Pt atoms per nm²), this resulted in a macroscopic coverage of $2.22/14.99 = 14.80\%$. Furthermore, the force exerted by an adsorbed O atom on a Pt atom was determined from the RMD simulation to be $1.17 \times 10^{-10}$ N. The compressive stress applied on the edge surfaces of the nanocrystal was calculated to be $1.17 \times 10^{-10} \times 2.22 \times 10^{18} = 259.74$ MPa. Furthermore, only a fraction of the adsorbed oxygen atoms was an active site for final product formation owing to steric hindrance. We calculated the ratio of the adsorbed methane to adsorbed oxygen atoms on the edges and corners of the Pt nanocrystal from the RMD simulation (Fig. 4a) to be $0.398 \pm 0.003$. This yielded a methane adsorption activity per unit area of 0.88 atoms per nm² and coverage on the Pt nanocrystal of 5.88%. The additional force on a Pt atom from methane adsorption was found from RMD to span (averaged over at least 50 ps of simulation before and after the methane adsorption event)

$0.49 \times 10^{-10}$–$1.84 \times 10^{-10}$ N with a mean of $1.12 \times 10^{-10}$ N. The additional compressive stress on the nanocrystal was, thus, taken to be 98.79 MPa. Within typical RMD timescales, there were not enough final product formation events to obtain a good statistical estimate of the additional force. We used the obtained mean of $1.12 \times 10^{-10}$ N in the FEA simulation. Over the entire span of the observed RMD force, the FEA maximum [111] projected displacement lay between 0.05 and 0.08 nm.

**Data availability**. The data reported in this paper are available upon request. All code, including the reconstruction algorithm, is also available upon request.

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

## Acknowledgements

We thank S.B. Mun (GIST) and Y.H. Kim (KICET) for useful discussion for the catalytic processes and for initial sample preparation, respectively. This research was supported by the National Research Foundation of Korea (NRF-2014R1A2A1A10052454, 2015R1A5A1009962, 2016R1A6B2A02005468, and 2017K1A3A7A09016379). Use of the Advanced Photon Source and computational resources at the Center for Nanoscale Materials was supported by the US Department of Energy, Office of Science, Office of Basic Energy Science, under contract no. DE-AC02-06CH11357. This research used resources of the National Energy Research Scientific Computing Center, a DOE Office of Science User Facility supported by the Office of Science of the U.S. Department of Energy under contract no. DE-AC02-05CH11231. M.J.C. acknowledges funding from Argonne LDRD 2018-019-N0: A.I.C.D.I.: Atomistically Informed Coherent Diffraction Imaging.

## Author contributions

Coherent X-ray diffraction measurements were carried out by D.K., M.C., J.C., S.K., K.Y., J.K., W.C., E.M., R.H., A.Z., M.S., and H.K. CXD data analysis was carried out by D.K., M.C., J.C., and K.Y. Pt nanocrystal growth was carried out by D.K., M.C., J.C. with advice from D.R. Gas analysis measurements were carried out by D.K. and J.K. Reactive molecular dynamics and FEA calculation were carried out by M.J.C. and K.S. under supervision of S.K.R.S.S. H.K. conceived the experiment. D.K., W.C., M.J.C., R.H., K.S., and H.K. wrote the paper. All authors discussed the results and commented on the manuscript.

## Additional information

**Competing interests:** The authors declare no competing interests.

