## [Peer Review File · Nature Communications]

Reviewers' comments:

Reviewer #1 (Remarks to the Author):

This manuscript reports impressive work taking full advantage of the latest capabilities of modern synchrotron radiation facilities. The experiment reports the measurement of deformation of a Pt nanocrystal to show a large change of compressive strain at its edges indicating localization of active sites during catalytic methane oxidation. The result is confirmed with a Reactive molecular dynamics (RMD) simulation. It is also roughly what is expected from numerous previous studies on the AVERAGE materials, but not seen before on individual nanocrystals.

The work is thoroughly analysed and as complete as possible given the current measurement capabilities. It should be published in Nature Communications.

Minor corrections only:

p3 The detailed experimental

p5 What are "dissociated O atoms"? Is it O₂ splitting? Please define.

Reviewer #2 (Remarks to the Author):

In this contribution, in-situ Bragg coherent diffraction imaging was carried out to investigate the distortions of Pt nanocrystals under different reaction atmospheres. Moreover, reactive molecular dynamics and finite element analysis were performed to simulate the deformation field distribution. All these came to the conclusion that the interaction between the reactants and the Pt nanocrystals would induce significant structural changes of nanocrystals, especially at the corner and edge region. Without any doubt, this work would pave an avenue to explore the dynamic structure of heterogeneous catalysts during the catalysis. I strongly recommend its publication in Nature Communications at present form.

Responses to Reviewers:

Reviewer #1 (Remarks to the Author):

This manuscript reports impressive work taking full advantage of the latest capabilities of modern synchrotron radiation facilities. The experiment reports the measurement of deformation of a Pt nanocrystal to show a large change of compressive strain at its edges indicating localization of active sites during catalytic methane oxidation. The result is confirmed with a Reactive molecular dynamics (RMD) simulation. It is also roughly what is expected from numerous previous studies on the AVERAGE materials, but not seen before on individual nanocrystals.

The work is thoroughly analysed and as complete as possible given the current measurement capabilities. It should be published in Nature Communications.

Minor corrections only:

p3 The detailed experimental

We fixed it. (In addition, we found a typo in line 14, p.4, we changed “show” to “shows”).

p5 What are "dissociated O atoms"? Is it O₂ splitting? Please define.

It is O atoms generated by O₂ dissociation based on the process, $O_2^{*+*} \rightarrow 2O^*$, in step 2.2 shown in Scheme 2 in Ref. 9.

Based on the referee's suggestion, we changed “dissociated O atoms” (in p.5) to “due to chemisorbed O atoms from dissociation of O₂ molecules”.